# Towards Principled Methods for Training Generative Adversarial Networks

**Martin Arjovsky**
Courant Institute of Mathematical Sciences
martinarjovsky@gmail.com

**Léon Bottou**
Facebook AI Research
leonb@fb.com

## Abstract

The goal of this paper is not to introduce a single algorithm or method, but to make theoretical steps towards fully understanding the training dynamics of generative adversarial networks. In order to substantiate our theoretical analysis, we perform targeted experiments to verify our assumptions, illustrate our claims, and quantify the phenomena. This paper is divided into three sections. The first section introduces the problem at hand. The second section is dedicated to studying and proving rigorously the problems including instability and saturation that arize when training generative adversarial networks. The third section examines a practical and theoretically grounded direction towards solving these problems, while introducing new tools to study them.

## 1 Introduction

Generative adversarial networks (GANs)(Goodfellow et al., 2014a) have achieved great success at generating realistic and sharp looking images. However, they are widely general methods, now starting to be applied to several other important problems, such as semisupervised learning, stabilizing sequence learning methods for speech and language, and 3D modelling. (Denton et al., 2015; Radford et al., 2015; Salimans et al., 2016; Lamb et al., 2016; Wu et al., 2016)

However, they still remain remarkably difficult to train, with most current papers dedicated to heuristically finding stable architectures. (Radford et al., 2015; Salimans et al., 2016)

Despite their success, there is little to no theory explaining the unstable behaviour of GAN training. Furthermore, approaches to attacking this problem still rely on heuristics that are extremely sensitive to modifications. This makes it extremely hard to experiment with new variants, or to use them in new domains, which limits their applicability drastically. This paper aims to change that, by providing a solid understanding of these issues, and creating principled research directions towards adressing them.

It is interesting to note that the architecture of the generator used by GANs doesn't differ significantly from other approaches like variational autoencoders (Kingma & Welling, 2013). After all, at the core of it we first sample from a simple prior $z \sim p(z)$, and then output our final sample $g_\theta(z)$, sometimes adding noise in the end. Always, $g_\theta$ is a neural network parameterized by $\theta$, and the main difference is how $g_\theta$ is trained.

Traditional approaches to generative modeling relied on maximizing likelihood, or equivalently minimizing the Kullback-Leibler (KL) divergence between our unknown data distribution $\mathbb{P}_r$ and our generator's distribution $\mathbb{P}_g$ (that depends of course on $\theta$). If we assume that both distributions are continuous with densities $P_r$ and $P_g$, then these methods try to minimize

$$KL(\mathbb{P}_r \| \mathbb{P}_g) = \int_{\mathcal{X}} P_r(x) \log \frac{P_r(x)}{P_g(x)} \, \mathrm{d}x$$

This cost function has the good property that it has a unique minimum at $\mathbb{P}_g = \mathbb{P}_r$, and it doesn't require knowledge of the unknown $P_r(x)$ to optimize it (only samples). However, it is interesting to see how this divergence is not symetrical between $\mathbb{P}_r$ and $\mathbb{P}_g$:

- If $P_r(x) > P_g(x)$, then $x$ is a point with higher probability of coming from the data than being a generated sample. This is the core of the phenomenon commonly described as 'mode dropping': when there are large regions with high values of $P_r$, but small or zero values in $P_g$. It is important to note that when $P_r(x) > 0$ but $P_g(x) \to 0$, the integrand inside the KL grows quickly to infinity, meaning that this cost function assigns an extremely high cost to a generator's distribution not covering parts of the data.

- If $P_r(x) < P_g(x)$, then $x$ has low probability of being a data point, but high probability of being generated by our model. This is the case when we see our generator outputting an image that doesn't look real. In this case, when $P_r(x) \to 0$ and $P_g(x) > 0$, we see that the value inside the KL goes to 0, meaning that this cost function will pay extremely low cost for generating fake looking samples.

Clearly, if we would minimize $KL(\mathbb{P}_g \| \mathbb{P}_r)$ instead, the weighting of these errors would be reversed, meaning that this cost function would pay a high cost for generating not plausibly looking pictures. Generative adversarial networks have been shown to optimize (in its original formulation), the Jensen-shannon divergence, a symmetric middle ground to this two cost functions

$$JSD(\mathbb{P}_r \| \mathbb{P}_g) = \frac{1}{2} KL(\mathbb{P}_r \| \mathbb{P}_A) + \frac{1}{2} KL(\mathbb{P}_g \| \mathbb{P}_A)$$

where $\mathbb{P}_A$ is the 'average' distribution, with density $\frac{P_r + P_g}{2}$. An impressive experimental analysis of the similarities, uses and differences of these divergences in practice can be seen at Theis et al. (2016). It is indeed conjectured that the reason of GANs success at producing realistically looking images is due to the switch from the traditional maximum likelihood approaches. (Theis et al., 2016; Huszar, 2015). However, the problem is far from closed.

Generative adversarial networks are formulated in two steps. We first train a discriminator $D$ to maximize

$$L(D, g_\theta) = \mathbb{E}_{x \sim \mathbb{P}_r}[\log D(x)] + \mathbb{E}_{x \sim \mathbb{P}_g}[\log(1 - D(x))] \tag{1}$$

One can show easily that the optimal discriminator has the shape

$$D^*(x) = \frac{P_r(x)}{P_r(x) + P_g(x)} \tag{2}$$

and that $L(D^*, g_\theta) = 2JSD(\mathbb{P}_r \| \mathbb{P}_g) - 2 \log 2$, so minimizing equation (1) as a function of $\theta$ yields minimizing the Jensen-Shannon divergence when the discriminator is optimal. In theory, one would expect therefore that we would first train the discriminator as close as we can to optimality (so the cost function on $\theta$ better approximates the $JSD$), and then do gradient steps on $\theta$, alternating these two things. However, this doesn't work. In practice, as the discriminator gets better, the updates to the generator get consistently worse. The original GAN paper argued that this issue arose from saturation, and switched to another similar cost function that doesn't have this problem. However, even with this new cost function, updates tend to get worse and optimization gets massively unstable. Therefore, several questions arize:

- Why do updates get worse as the discriminator gets better? Both in the original and the new cost function.

- Why is GAN training massively unstable?

- Is the new cost function following a similar divergence to the $JSD$? If so, what are its properties?

- Is there a way to avoid some of these issues?

The fundamental contributions of this paper are the answer to all these questions, and perhaps more importantly, to introduce the tools to analyze them properly. We provide a new direction designed to avoid the instability issues in GANs, and examine in depth the theory behind it. Finally, we state a series of open questions and problems, that determine several new directions of research that begin with our methods.

## 2 SOURCES OF INSTABILITY

The theory tells us that the trained discriminator will have cost at most $2 \log 2 - 2JSD(\mathbb{P}_r \| \mathbb{P}_g)$. However, in practice, if we just train $D$ till convergence, its error will go to 0, as observed in Figure 1, pointing to the fact that the $JSD$ between them is maxed out. The only way this can happen is if the distributions are not continuous[1], or they have disjoint supports.

One possible cause for the distributions not to be continuous is if their supports lie on low dimensional manifolds. There is strong empirical and theoretical evidence to believe that $\mathbb{P}_r$ is indeed extremely concentrated on a low dimensional manifold (Narayanan & Mitter, 2010). As of $\mathbb{P}_g$, we will prove soon that such is the case as well.

In the case of GANs, $\mathbb{P}_g$ is defined via sampling from a simple prior $z \sim p(z)$, and then applying a function $g : \mathcal{Z} \to \mathcal{X}$, so the support of $\mathbb{P}_g$ has to be contained in $g(\mathcal{Z})$. If the dimensionality of $\mathcal{Z}$ is less than the dimension of $\mathcal{X}$ (as is typically the case), then it's imposible for $\mathbb{P}_g$ to be continuous. This is because in most cases $g(\mathcal{Z})$ will be contained in a union of low dimensional manifolds, and therefore have measure 0 in $\mathcal{X}$. Note that while intuitive, this is highly nontrivial, since having an $n$-dimensional parameterization does absolutely not imply that the image will lie on an $n$-dimensional manifold. In fact, there are many easy counterexamples, such as Peano curves, lemniscates, and many more. In order to show this for our case, we rely heavily on $g$ being a neural network, since we are able to leverage that $g$ is made by composing very well behaved functions. We now state this properly in the following Lemma:

**Lemma 1.** *Let $g : \mathcal{Z} \to \mathcal{X}$ be a function composed by affine transformations and pointwise nonlinearities, which can either be rectifiers, leaky rectifiers, or smooth strictly increasing functions (such as the sigmoid, tanh, softplus, etc). Then, $g(\mathcal{Z})$ is contained in a countable union of manifolds of dimension at most $\dim \mathcal{Z}$. Therefore, if the dimension of $\mathcal{Z}$ is less than the one of $\mathcal{X}$, $g(\mathcal{Z})$ will be a set of measure 0 in $\mathcal{X}$.*

*Proof.* See Appendix A. □

Driven by this, this section shows that if the supports of $\mathbb{P}_r$ and $\mathbb{P}_g$ are disjoint or lie in low dimensional manifolds, there is always a perfect discriminator between them, and we explain exactly how and why this leads to an unreliable training of the generator.

### 2.1 THE PERFECT DISCRIMINATION THEOREMS

For simplicity, and to introduce the methods, we will first explain the case where $\mathbb{P}_r$ and $\mathbb{P}_g$ have disjoint supports. We say that a discriminator $D : \mathcal{X} \to [0, 1]$ has accuracy 1 if it takes the value 1 on a set that contains the support of $\mathbb{P}_r$ and value 0 on a set that contains the support of $\mathbb{P}_g$. Namely, $\mathbb{P}_r[D(x) = 1] = 1$ and $\mathbb{P}_g[D(x) = 0] = 1$.

**Theorem 2.1.** *If two distributions $\mathbb{P}_r$ and $\mathbb{P}_g$ have support contained on two disjoint compact subsets $\mathcal{M}$ and $\mathcal{P}$ respectively, then there is a smooth optimal discrimator $D^* : \mathcal{X} \to [0, 1]$ that has accuracy 1 and $\nabla_x D^*(x) = 0$ for all $x \in \mathcal{M} \cup \mathcal{P}$.*

*Proof.* The discriminator is trained to maximize

$$\mathbb{E}_{x \sim \mathbb{P}_r}[\log D(x)] + \mathbb{E}_{x \sim \mathbb{P}_g}[\log(1 - D(x))]$$

Since $\mathcal{M}$ and $\mathcal{P}$ are compact and disjoint, $0 < \delta = d(\mathcal{P}, \mathcal{M})$ the distance between both sets. We now define

$$\hat{\mathcal{M}} = \{x : d(x, M) \le \delta/3\}$$
$$\hat{\mathcal{P}} = \{x : d(x, P) \le \delta/3\}$$

By definition of $\delta$ we have that $\hat{P}$ and $\hat{M}$ are clearly disjoint compact sets. Therefore, by Urysohn's smooth lemma there exists a smooth function $D^* : \mathcal{X} \to [0, 1]$ such that $D^*|_{\hat{\mathcal{M}}} \equiv 1$ and $D^*|_{\hat{\mathcal{P}}} \equiv 0$. Since $\log D^*(x) = 0$ for all $x$ in the support of $\mathbb{P}_r$ and $\log(1 - D^*(x)) = 0$ for all $x$ in the support of $\mathbb{P}_g$, the discriminator is completely optimal and has accuracy 1. Furthermore, let $x$ be in $\mathcal{M} \cup \mathcal{P}$. If we assume that $x \in \mathcal{M}$, there is an open ball $B = B(x, \delta/3)$ on which $D^*|_B$ is constant. This shows that $\nabla_x D^*(x) \equiv 0$. Taking $x \in \mathcal{P}$ and working analogously we finish the proof. □

---

[1] By continuous we will actually refer to an absolutely continuous random variable (i.e. one that has a density), as it typically done. For further clarification see Appendix B.

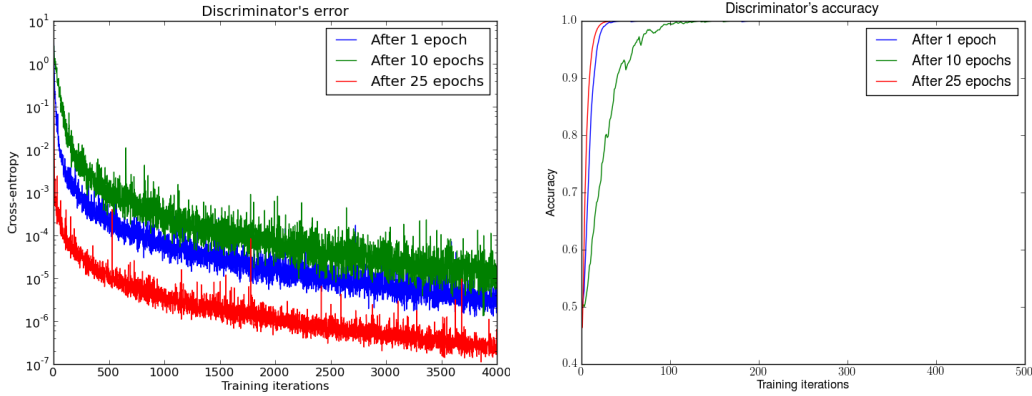

Figure 1: First, we trained a DCGAN for 1, 10 and 25 epochs. Then, with the generator fixed we train a discriminator from scratch. We see the error quickly going to 0, even with very few iterations on the discriminator. This even happens after 25 epochs of the DCGAN, when the samples are remarkably good and the supports are likely to intersect, pointing to the non-continuity of the distributions. Note the logarithmic scale. For illustration purposes we also show the accuracy of the discriminator, which goes to 1 in sometimes less than 50 iterations. This is 1 even for numerical precision, and the numbers are running averages, pointing towards even faster convergence.

In the next theorem, we take away the disjoint assumption, to make it general to the case of two different manifolds. However, if the two manifolds match perfectly on a big part of the space, no discriminator could separate them. Intuitively, the chances of two low dimensional manifolds having this property is rather dim: for two curves to match in space in a specific segment, they couldn't be perturbed in any arbitrarily small way and still satisfy this property. To do this, we will define the notion of two manifolds perfectly aligning, and show that this property never holds with probability 1 under any arbitrarily small perturbations.

**Definition 2.1.** We first need to recall the definition of transversallity. Let $\mathcal{M}$ and $\mathcal{P}$ be two boundary free regular submanifolds of $\mathcal{F}$, which in our cases will simply be $\mathcal{F} = \mathbb{R}^d$. Let $x \in \mathcal{M} \cap \mathcal{P}$ be an intersection point of the two manifolds. We say that $\mathcal{M}$ and $\mathcal{P}$ intersect transversally in $x$ if $T_x\mathcal{M} + T_x\mathcal{P} = T_x\mathcal{F}$, where $T_x\mathcal{M}$ means the tangent space of $\mathcal{M}$ around $x$.

**Definition 2.2.** We say that two manifolds without boundary $\mathcal{M}$ and $\mathcal{P}$ **perfectly align** if there is an $x \in \mathcal{M} \cap \mathcal{P}$ such that $\mathcal{M}$ and $\mathcal{P}$ don't intersect transversally in $x$.
We shall note the boundary and interior of a manifold $\mathcal{M}$ by $\partial M$ and Int $M$ respectively. We say that two manifolds $\mathcal{M}$ and $\mathcal{P}$ (with or without boundary) perfectly align if any of the boundary free manifold pairs (Int $\mathcal{M}$, Int $\mathcal{P}$), (Int $\mathcal{M}$, $\partial\mathcal{P}$), ($\partial\mathcal{M}$, Int $\mathcal{P}$) or ($\partial\mathcal{M}$, $\partial\mathcal{P}$) perfectly align.

The interesting thing is that we can safely assume in practice that any two manifolds never perfectly align. This can be done since an arbitrarilly small random perturbation on two manifolds will lead them to intersect transversally or don't intersect at all. This is precisely stated and proven in Lemma 2.

As stated by Lemma 3, if two manifolds don't perfectly align, their intersection $\mathcal{L} = \mathcal{M} \cap \mathcal{P}$ will be a finite union of manifolds with dimensions strictly lower than both the dimension of $\mathcal{M}$ and the one of $\mathcal{P}$.

**Lemma 2.** *Let $\mathcal{M}$ and $\mathcal{P}$ be two regular submanifolds of $\mathbb{R}^d$ that don't have full dimension. Let $\eta, \eta'$ be arbitrary independent continuous random variables. We therefore define the perturbed manifolds as $\tilde{\mathcal{M}} = \mathcal{M} + \eta$ and $\tilde{\mathcal{P}} = \mathcal{P} + \eta'$. Then*

$$\mathbb{P}_{\eta,\eta'}(\tilde{\mathcal{M}} \text{ does not perfectly align with } \tilde{\mathcal{P}}) = 1$$

*Proof.* See Appendix A. □

**Lemma 3.** *Let $\mathcal{M}$ and $\mathcal{P}$ be two regular submanifolds of $\mathbb{R}^d$ that don't perfectly align and don't have full dimension. Let $\mathcal{L} = \mathcal{M} \cap \mathcal{P}$. If $\mathcal{M}$ and $\mathcal{P}$ don't have boundary, then $\mathcal{L}$ is also a manifold, and has strictly lower dimension than both the one of $\mathcal{M}$ and the one of $\mathcal{P}$. If they have boundary, $\mathcal{L}$ is a union of at most 4 strictly lower dimensional manifolds. In both cases, $\mathcal{L}$ has measure 0 in both $\mathcal{M}$ and $\mathcal{P}$.*

*Proof.* See Appendix A. □

We now state our perfect discrimination result for the case of two manifolds.

**Theorem 2.2.** *Let $\mathbb{P}_r$ and $\mathbb{P}_g$ be two distributions that have support contained in two closed manifolds $\mathcal{M}$ and $\mathcal{P}$ that don't perfectly align and don't have full dimension. We further assume that $\mathbb{P}_r$ and $\mathbb{P}_g$ are continuous in their respective manifolds, meaning that if there is a set $A$ with measure 0 in $\mathcal{M}$, then $\mathbb{P}_r(A) = 0$ (and analogously for $\mathbb{P}_g$). Then, there exists an optimal discriminator $D^* : \mathcal{X} \to [0, 1]$ that has accuracy 1 and for almost any $x$ in $\mathcal{M}$ or $\mathcal{P}$, $D^*$ is smooth in a neighbourhood of $x$ and $\nabla_x D^*(x) = 0$.*

*Proof.* By Lemma 3 we know that $\mathcal{L} = \mathcal{M} \cap \mathcal{P}$ is strictly lower dimensional than both $\mathcal{M}$ and $\mathcal{P}$, and has measure 0 on both of them. By continuity, $\mathbb{P}_r(\mathcal{L}) = 0$ and $\mathbb{P}_g(\mathcal{L}) = 0$. Note that this implies the support of $\mathbb{P}_r$ is contained in $\mathcal{M} \setminus \mathcal{L}$ and the support of $\mathbb{P}_g$ is contained in $\mathcal{P} \setminus \mathcal{L}$.

Let $x \in \mathcal{M} \setminus \mathcal{L}$. Therefore, $x \in \mathcal{P}^c$ (the complement of $\mathcal{P}$) which is an open set, so there exists a ball of radius $\epsilon_x$ such that $B(x, \epsilon_x) \cap \mathcal{P} = \emptyset$. This way, we define

$$\hat{\mathcal{M}} = \bigcup_{x \in \mathcal{M} \setminus \mathcal{L}} B(x, \epsilon_x/3)$$

We define $\hat{\mathcal{P}}$ analogously. Note that by construction these are both open sets on $\mathbb{R}^d$. Since $\mathcal{M} \setminus \mathcal{L} \subseteq \hat{\mathcal{M}}$, and $\mathcal{P} \setminus \mathcal{L} \subseteq \hat{\mathcal{P}}$, the support of $\mathbb{P}_r$ and $\mathbb{P}_g$ is contained in $\hat{\mathcal{M}}$ and $\hat{\mathcal{P}}$ respectively. As well by construction, $\hat{\mathcal{M}} \cap \hat{\mathcal{P}} = \emptyset$.

Let us define $D^*(x) = 1$ for all $x \in \hat{\mathcal{M}}$, and 0 elsewhere (clearly including $\hat{\mathcal{P}}$. Since $\log D^*(x) = 0$ for all $x$ in the support of $\mathbb{P}_r$ and $\log(1 - D^*(x)) = 0$ for all $x$ in the support of $\mathbb{P}_g$, the discriminator is completely optimal and has accuracy 1. Furthermore, let $x \in \hat{\mathcal{M}}$. Since $\hat{\mathcal{M}}$ is an open set and $D^*$ is constant on $\hat{\mathcal{M}}$, then $\nabla_x D^*|_{\hat{\mathcal{M}}} \equiv 0$. Analogously, $\nabla_x D^*|_{\hat{\mathcal{P}}} \equiv 0$. Therefore, the set of points where $D^*$ is non-smooth or has non-zero gradient inside $\mathcal{M} \cup \mathcal{P}$ is contained in $\mathcal{L}$, which has null-measure in both manifolds, therefore concluding the theorem. □

These two theorems tell us that there are perfect discriminators which are smooth and constant almost everywhere in $\mathcal{M}$ and $\mathcal{P}$. The fact that the discriminator is constant in both manifolds points to the fact that we won't really be able to learn anything by backproping through it, as we shall see in the next subsection. To conclude this general statement, we state the following theorem on the divergences of $\mathbb{P}_r$ and $\mathbb{P}_g$, whose proof is trivial and left as an exercise to the reader.

**Theorem 2.3.** *Let $\mathbb{P}_r$ and $\mathbb{P}_g$ be two distributions whose support lies in two manifolds $\mathcal{M}$ and $\mathcal{P}$ that don't have full dimension and don't perfectly align. We further assume that $\mathbb{P}_r$ and $\mathbb{P}_g$ are continuous in their respective manifolds. Then,*

$$JSD(\mathbb{P}_r \| \mathbb{P}_g) = \log 2$$
$$KL(\mathbb{P}_r \| \mathbb{P}_g) = +\infty$$
$$KL(\mathbb{P}_g \| \mathbb{P}_r) = +\infty$$

Note that these divergences will be maxed out even if the two manifolds lie arbitrarilly close to each other. The samples of our generator might look impressively good, yet both KL divergences will be infinity. Therefore, Theorem 2.3 points us to the fact that attempting to use divergences out of the box to test similarities between the distributions we typically consider might be a terrible idea. Needless to say, if these divergencies are always maxed out attempting to minimize them by gradient descent isn't really possible. We would like to have a perhaps softer measure, that incorporates a notion of distance between the points in the manifolds. We will come back to this topic later in section 3, where we explain an alternative metric and provide bounds on it that we are able to analyze and optimize.

## 2.2 THE CONSEQUENCES, AND THE PROBLEMS OF EACH COST FUNCTION

Theorems 2.1 and 2.2 showed one very important fact. If the two distributions we care about have supports that are disjoint or lie on low dimensional manifolds, the optimal discriminator will be perfect and its gradient will be zero almost everywhere.

### 2.2.1 THE ORIGINAL COST FUNCTION

We will now explore what happens when we pass gradients to the generator through a discriminator. One crucial difference with the typical analysis done so far is that we will develop the theory for an **approximation** to the optimal discriminator, instead of working with the (unknown) true discriminator. We will prove that as the approximaton gets better, either we see vanishing gradients or the massively unstable behaviour we see in practice, depending on which cost function we use.

In what follows, we denote by $\|D\|$ the norm

$$\|D\| = \sup_{x \in \mathcal{X}} |D(x)| + \|\nabla_x D(x)\|_2$$

The use of this norm is to make the proofs simpler, but could have been done in another Sobolev norm $\| \cdot \|_{1,p}$ for $p < \infty$ covered by the universal approximation theorem in the sense that we can guarantee a neural network approximation in this norm (Hornik, 1991).

**Theorem 2.4** (**Vanishing gradients on the generator**). *Let $g_\theta : \mathcal{Z} \to \mathcal{X}$ be a differentiable function that induces a distribution $\mathbb{P}_g$. Let $\mathbb{P}_r$ be the real data distribution. Let $D$ be a differentiable discriminator. If the conditions of Theorems 2.1 or 2.2 are satisfied, $\|D - D^*\| < \epsilon$, and $\mathbb{E}_{z \sim p(z)} \left[ \|J_\theta g_\theta(z)\|_2^2 \right] \leq M^2$, [2] then*

$$\|\nabla_\theta \mathbb{E}_{z \sim p(z)}[\log(1 - D(g_\theta(z)))]\|_2 < M \frac{\epsilon}{1 - \epsilon}$$

*Proof.* In both proofs of Theorems 2.1 and 2.2 we showed that $D^*$ is locally 0 on the support of $\mathbb{P}_g$. Then, using Jensen's inequality and the chain rule on this support we have

$$
\begin{aligned}
\|\nabla_\theta \mathbb{E}_{z \sim p(z)}[\log(1 - D(g_\theta(z)))]\|_2^2 &\leq \mathbb{E}_{z \sim p(z)} \left[ \frac{\|\nabla_\theta D(g_\theta(z))\|_2^2}{|1 - D(g_\theta(z))|^2} \right] \\
&\leq \mathbb{E}_{z \sim p(z)} \left[ \frac{\|\nabla_x D(g_\theta(z))\|_2^2 \|J_\theta g_\theta(z)\|_2^2}{|1 - D(g_\theta(z))|^2} \right] \\
&< \mathbb{E}_{z \sim p(z)} \left[ \frac{\left( \|\nabla_x D^*(g_\theta(z))\|_2 + \epsilon \right)^2 \|J_\theta g_\theta(z)\|_2^2}{\left( |1 - D^*(g_\theta(z))| - \epsilon \right)^2} \right] \\
&= \mathbb{E}_{z \sim p(z)} \left[ \frac{\epsilon^2 \|J_\theta g_\theta(z)\|_2^2}{(1 - \epsilon)^2} \right] \\
&\leq M^2 \frac{\epsilon^2}{(1 - \epsilon)^2}
\end{aligned}
$$

Taking square root of everything we get

$$\|\nabla_\theta \mathbb{E}_{z \sim p(z)}[\log(1 - D(g_\theta(z)))]\|_2 < M \frac{\epsilon}{1 - \epsilon}$$

finishing the proof $\qquad \square$

**Corollary 2.1.** *Under the same assumptions of Theorem 2.4*

$$\lim_{\|D - D^*\| \to 0} \nabla_\theta \mathbb{E}_{z \sim p(z)}[\log(1 - D(g_\theta(z)))] = 0$$

---

[2]Since $M$ can depend on $\theta$, this condition is trivially verified for a uniform prior and a neural network. The case of a Gaussian prior requires more work because we need to bound the growth on $z$, but is also true for current architectures.

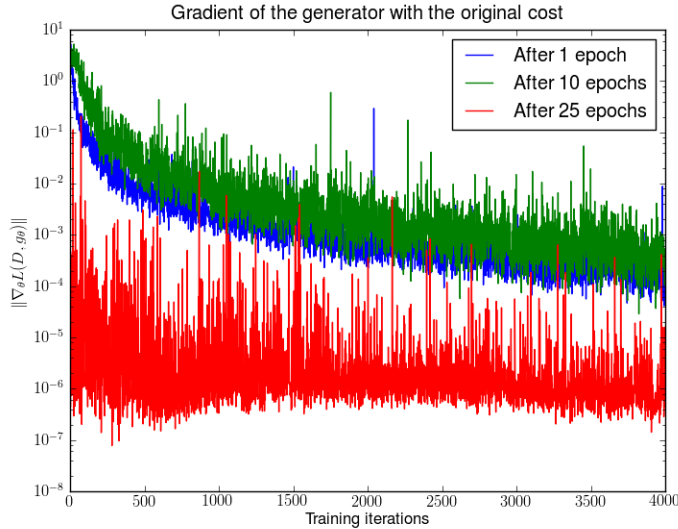

Figure 2: First, we trained a DCGAN for 1, 10 and 25 epochs. Then, with the generator fixed we train a discriminator from scratch and measure the gradients with the original cost function. We see the gradient norms decay quickly, in the best case 5 orders of magnitude after 4000 discriminator iterations. Note the logarithmic scale.

This shows that as our discriminator gets better, the gradient of the generator vanishes. For completeness, this was experimentally verified in Figure 2. The fact that this happens is terrible, since the fact that the generator's cost function being close to the Jensen Shannon divergence depends on the quality of this approximation. This points us to a fundamental: either our updates to the discriminator will be inacurate, or they will vanish. This makes it difficult to train using this cost function, or leave up to the user to decide the precise amount of training dedicated to the discriminator, which can make GAN training extremely hard.

### 2.2.2   THE $-\log D$ ALTERNATIVE

To avoid gradients vanishing when the discriminator is very confident, people have chosen to use a different gradient step for the generator.

$$\Delta\theta = \nabla_\theta \mathbb{E}_{z\sim p(z)}\left[-\log D(g_\theta(z))\right]$$

We now state and prove for the first time which cost function is being optimized by this gradient step. Later, we prove that while this gradient doesn't necessarily suffer from vanishing gradients, it does cause massively unstable updates (that have been widely experienced in practice) under the prescence of a noisy approximation to the optimal discriminator.

**Theorem 2.5.** *Let $\mathbb{P}_r$ and $\mathbb{P}_{g_\theta}$ be two continuous distributions, with densities $P_r$ and $P_{g_\theta}$ respectively. Let $D^* = \frac{P_r}{P_{g_{\theta_0}}+P_r}$ be the optimal discriminator, fixed for a value $\theta_0$[3]. Therefore,*

$$\mathbb{E}_{z\sim p(z)}\left[-\nabla_\theta \log D^*(g_\theta(z))|_{\theta=\theta_0}\right] = \nabla_\theta\left[KL(\mathbb{P}_{g_\theta}\|\mathbb{P}_r) - 2JSD(\mathbb{P}_{g_\theta}\|\mathbb{P}_r)\right]|_{\theta=\theta_0} \qquad (3)$$

Before diving into the proof, let's look at equation (3) for a second. This is the inverted KL minus two JSD. First of all, the JSDs are in the opposite sign, which means they are pushing for the distributions to be different, which seems like a fault in the update. Second, the KL appearing in the equation is $KL(\mathbb{P}_g\|\mathbb{P}_r)$, not the one equivalent to maximum likelihood. As we know, this KL assigns an extremely high cost to generating fake looking samples, and an extremely low cost on mode dropping; and the JSD is symetrical so it shouldn't alter this behaviour. This explains what we see in practice, that GANs (when stabilized) create good looking samples, and justifies what is commonly conjectured, that GANs suffer from an extensive amount of mode dropping.

---

[3]This is important since when backpropagating to the generator, the discriminator is assumed fixed

*Proof.* We already know by Goodfellow et al. (2014a) that

$$\mathbb{E}_{z \sim p(z)} \left[ \nabla_\theta \log(1 - D^*(g_\theta(z)))|_{\theta=\theta_0} \right] = \nabla_\theta 2JSD(\mathbb{P}_{g_\theta} \| \mathbb{P}_r)|_{\theta=\theta_0}$$

Furthermore, as remarked by Huszar (2016),

$$
\begin{aligned}
KL(\mathbb{P}_{g_\theta} \| \mathbb{P}_r) &= \mathbb{E}_{x \sim \mathbb{P}_{g_\theta}} \left[ \log \frac{P_{g_\theta}(x)}{P_r(x)} \right] \\
&= \mathbb{E}_{x \sim \mathbb{P}_{g_\theta}} \left[ \log \frac{P_{g_{\theta_0}}(x)}{P_r(x)} \right] - \mathbb{E}_{x \sim \mathbb{P}_{g_\theta}} \left[ \log \frac{P_{g_\theta}(x)}{P_{g_{\theta_0}}(x)} \right] \\
&= -\mathbb{E}_{x \sim \mathbb{P}_{g_\theta}} \left[ \log \frac{D^*(x)}{1 - D^*(x)} \right] - KL(\mathbb{P}_{g_\theta} \| \mathbb{P}_{g_{\theta_0}}) \\
&= -\mathbb{E}_{z \sim p(z)} \left[ \log \frac{D^*(g_\theta(z))}{1 - D^*(g_\theta(z))} \right] - KL(\mathbb{P}_{g_\theta} \| \mathbb{P}_{g_{\theta_0}})
\end{aligned}
$$

Taking derivatives in $\theta$ at $\theta_0$ we get

$$
\begin{aligned}
\nabla_\theta KL(\mathbb{P}_{g_\theta} \| \mathbb{P}_r)|_{\theta=\theta_0} &= -\nabla_\theta \mathbb{E}_{z \sim p(z)} \left[ \log \frac{D^*(g_\theta(z))}{1 - D^*(g_\theta(z))} \right] |_{\theta=\theta_0} - \nabla_\theta KL(\mathbb{P}_{g_\theta} \| \mathbb{P}_{g_{\theta_0}})|_{\theta=\theta_0} \\
&= \mathbb{E}_{z \sim p(z)} \left[ -\nabla_\theta \log \frac{D^*(g_\theta(z))}{1 - D^*(g_\theta(z))} \right] |_{\theta=\theta_0}
\end{aligned}
$$

Substracting this last equation with the result for the JSD, we obtain our desired result. $\qquad\square$

We now turn to our result regarding the instability of a noisy version of the true distriminator.

**Theorem 2.6 (Instability of generator gradient updates).** *Let $g_\theta : \mathcal{Z} \to \mathcal{X}$ be a differentiable function that induces a distribution $\mathbb{P}_g$. Let $\mathbb{P}_r$ be the real data distribution, with either conditions of Theorems 2.1 or 2.2 satisfied. Let $D$ be a discriminator such that $D^* - D = \epsilon$ is a centered Gaussian process indexed by $x$ and independent for every $x$ (popularly known as white noise) and $\nabla_x D^* - \nabla_x D = r$ another independent centered Gaussian process indexed by $x$ and independent for every $x$. Then, each coordinate of*

$$\mathbb{E}_{z \sim p(z)} \left[ -\nabla_\theta \log D(g_\theta(z)) \right]$$

*is a centered Cauchy distribution with **infinite expectation and variance**.*[4]

*Proof.* Let us remember again that in this case $D$ is locally constant equal to 0 on the support of $\mathbb{P}_g$. We denote $r(z), \epsilon(z)$ the random variables $r(g_\theta(z)), \epsilon(g_\theta(z))$. By the chain rule and the definition of $r, \epsilon$, we get

$$
\begin{aligned}
\mathbb{E}_{z \sim p(z)} \left[ -\nabla_\theta \log D(g_\theta(z)) \right] &= \mathbb{E}_{z \sim p(z)} \left[ -\frac{J_\theta g_\theta(z) \nabla_x D(g_\theta(z))}{D(g_\theta(z))} \right] \\
&= \mathbb{E}_{z \sim p(z)} \left[ -\frac{J_\theta g_\theta(z) r(z)}{\epsilon(z)} \right]
\end{aligned}
$$

Since $r(z)$ is a centered Gaussian distribution, multiplying by a matrix doesn't change this fact. Furthermore, when we divide by $\epsilon(z)$, a centered Gaussian independent from the numerator, we get a centered Cauchy random variable on every coordinate. Averaging over $z$ the different independent Cauchy random variables again yields a centered Cauchy distribution. [5] $\qquad\square$

---

[4]Note that the theorem holds regardless of the variance of $r$ and $\epsilon$. As the approximation gets better, this error looks more and more as centered random noise due to the finite precision.

[5]A note on technicality: when $\epsilon$ is defined as such, the remaining process is not measurable in $x$, so we can't take the expectation in $z$ trivially. This is commonly bypassed, and can be formally worked out by stating the expectation as the result of a stochastic differential equation.

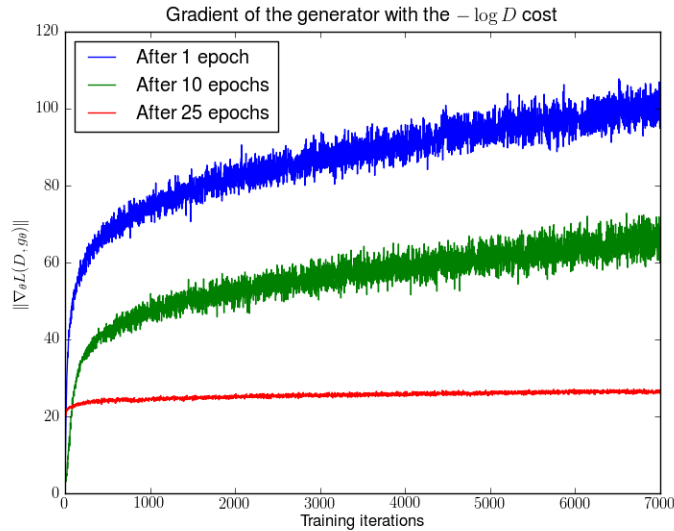

Figure 3: First, we trained a DCGAN for 1, 10 and 25 epochs. Then, with the generator fixed we train a discriminator from scratch and measure the gradients with the $-\log D$ cost function. We see the gradient norms grow quickly. Furthermore, the noise in the curves shows that the variance of the gradients is also increasing. All these gradients lead to updates that lower sample quality notoriously.

Note that even if we ignore the fact that the updates have infinite variance, we still arrive to the fact that the distribution of the updates is centered, meaning that if we bound the updates the expected update will be 0, providing no feedback to the gradient.

Since the assumption that the noises of $D$ and $\nabla D$ are decorrelated is albeit too strong, we show in Figure 3 how the norm of the gradient grows drastically as we train the discriminator closer to optimality, at any stage in training of a well stabilized DCGAN except when it has already converged. In all cases, using this updates lead to a notorious decrease in sample quality. The noise in the curves also shows that the variance of the gradients is increasing, which is known to delve into slower convergence and more unstable behaviour in the optimization (Bottou et al., 2016).

## 3 TOWARDS SOFTER METRICS AND DISTRIBUTIONS

An important question now is how to fix the instability and vanishing gradients issues. Something we can do to break the assumptions of these theorems is add continuous noise to the inputs of the discriminator, therefore smoothening the distribution of the probability mass.

**Theorem 3.1.** *If $X$ has distribution $\mathbb{P}_X$ with support on $\mathcal{M}$ and $\epsilon$ is an aboslutely continuous distribution with density $P_\epsilon$, then $\mathbb{P}_{X+\epsilon}$ is absolutely continuous with density*

$$P_{X+\epsilon}(x) = \mathbb{E}_{y \sim \mathbb{P}_X}\left[P_\epsilon(x-y)\right]$$

$$= \int_{\mathcal{M}} P_\epsilon(x-y)\, d\mathbb{P}_X(y)$$

*Proof.* See Appendix A. □

**Corollary 3.1.**  • *If $\epsilon \sim \mathcal{N}(0, \sigma^2 I)$ then*

$$P_{X+\epsilon}(x) = \frac{1}{Z}\int_{\mathcal{M}} e^{-\frac{\|y-x\|^2}{2\sigma^2}}\, d\mathbb{P}_X(y)$$

• *If $\epsilon \sim \mathcal{N}(0, \Sigma)$ then*

$$P_{X+\epsilon}(x) = \frac{1}{Z}\mathbb{E}_{y \sim \mathbb{P}_X}\left[e^{-\frac{1}{2}\|y-x\|_{\Sigma^{-1}}^2}\right]$$

- *If $P_\epsilon(x) \propto \frac{1}{\|x\|^{d+1}}$ then*

$$P_{X+\epsilon}(x) = \frac{1}{Z}\mathbb{E}_{y\sim\mathbb{P}_X}\left[\frac{1}{\|x-y\|^{d+1}}\right]$$

This theorem therefore tells us that the density $P_{X+\epsilon}(x)$ **is inversely proportional to the average distance to points in the support of $\mathbb{P}_X$, weighted by the probability of these points**. In the case of the support of $\mathbb{P}_X$ being a manifold, we will have the weighted average of the distance to the points along the manifold. How we choose the distribution of the noise $\epsilon$ will impact the notion of distance we are choosing. In our corolary, for example, we can see the effect of changing the covariance matrix by altering the norm inside the exponential. Different noises with different types of decays can therefore be used.

Now, the optimal discriminator between $\mathbb{P}_{g+\epsilon}$ and $\mathbb{P}_{r+\epsilon}$ is

$$D^*(x) = \frac{P_{r+\epsilon}(x)}{P_{r+\epsilon}(x) + P_{g+\epsilon}(x)}$$

and we want to calculate what the gradient passed to the generator is.

**Theorem 3.2.** *Let $\mathbb{P}_r$ and $\mathbb{P}_g$ be two distributions with support on $\mathcal{M}$ and $\mathcal{P}$ respectively, with $\epsilon \sim \mathcal{N}(0, \sigma^2 I)$. Then, the gradient passed to the generator has the form*

$$\mathbb{E}_{z\sim p(z)}\left[\nabla_\theta \log(1 - D^*(g_\theta(z)))\right] \tag{4}$$
$$= \mathbb{E}_{z\sim p(z)}\left[a(z)\int_\mathcal{M} P_\epsilon(g_\theta(z) - y)\nabla_\theta\|g_\theta(z) - y\|^2 \, \mathrm{d}\mathbb{P}_r(y)\right.$$
$$\left. - b(z)\int_\mathcal{P} P_\epsilon(g_\theta(z) - y)\nabla_\theta\|g_\theta(z) - y\|^2 \, \mathrm{d}\mathbb{P}_g(y)\right]$$

*where $a(z)$ and $b(z)$ are positive functions. Furthermore, $b > a$ if and only if $P_{r+\epsilon} > P_{g+\epsilon}$, and $b < a$ if and only if $P_{r+\epsilon} < P_{g+\epsilon}$.*

This theorem proves that we will drive our samples $g_\theta(z)$ **towards points along the data manifold, weighted by their probability and the distance from our samples**. Furthermore, the second term drives our points away from high probability samples, again, weighted by the sample manifold and distance to these samples. This is similar in spirit to contrastive divergence, where we lower the free energy of our samples and increase the free energy of data points. The importance of this term is seen more clearly when we have samples that have higher probability of coming from $\mathbb{P}_g$ than from $\mathbb{P}_r$. In this case, we will have $b > a$ and the second term will have the strength to lower the probability of this too likely samples. Finally, if there's an area around $x$ that has the same probability to come from $\mathbb{P}_g$ than $\mathbb{P}_r$, the gradient contributions between the two terms will cancel, therefore stabilizing the gradient when $\mathbb{P}_r$ is similar to $\mathbb{P}_g$.

There is one important problem with taking gradient steps exactly of the form (4), which is that in that case, $D$ will disregards errors that lie exactly in $g(\mathcal{Z})$, since this is a set of measure 0. However, $g$ will be optimizing its cost only on that space. This will make the discriminator extremely susceptible to adversarial examples, and will render low cost on the generator without high cost on the discriminator, and lousy meaningless samples. This is easily seen when we realize the term inside the expectation of equation (4) will be a positive scalar times $\nabla_x \log(1 - D^*(x))\nabla_\theta g_\theta(z)$, which is the directional derivative towards the exact adversarial term of Goodfellow et al. (2014b). Because of this, it is important to backprop through noisy samples in the generator as well. This will yield a crucial benefit: the generator's backprop term will be through samples on a set of positive measure that the discriminator will care about. Formalizing this notion, the actual gradient through the generator will now be proportional to $\nabla_\theta JSD(\mathbb{P}_{r+\epsilon}\|\mathbb{P}_{g+\epsilon})$, which will make the two noisy distributions match. As we anneal the noise, this will make $\mathbb{P}_r$ and $\mathbb{P}_g$ match as well. For completeness, we show the smooth gradient we get in this case. The proof is identical to the one of Theorem 3.2, so we leave it to the reader.

**Corollary 3.2.** *Let $\epsilon, \epsilon' \sim \mathcal{N}(0, \sigma^2 I)$ and $\tilde{g}_\theta(z) = g_\theta(z) + \epsilon'$, then*

$$\mathbb{E}_{z \sim p(z), \epsilon'} \left[ \nabla_\theta \log(1 - D^*(\tilde{g}_\theta(z))) \right] \tag{5}$$

$$= \mathbb{E}_{z \sim p(z), \epsilon'} \left[ a(z) \int_{\mathcal{M}} P_\epsilon(\tilde{g}_\theta(z) - y) \nabla_\theta \|\tilde{g}_\theta(z) - y\|^2 \, \mathrm{d}\mathbb{P}_r(y) \right.$$

$$\left. - b(z) \int_{\mathcal{P}} P_\epsilon(\tilde{g}_\theta(z) - y) \nabla_\theta \|\tilde{g}_\theta(z) - y\|^2 \, \mathrm{d}\mathbb{P}_g(y) \right]$$

$$= 2 \nabla_\theta JSD(\mathbb{P}_{r+\epsilon} \| \mathbb{P}_{g+\epsilon})$$

In the same as with Theorem 3.2, $a$ and $b$ will have the same properties. The main difference is that we will be moving all our noisy samples towards the data manifold, which can be thought of as moving a small neighbourhood of samples towards it. This will protect the discriminator against measure 0 adversarial examples.

*Proof of theorem 3.2.* Since the discriminator is assumed fixed when backproping to the generator, the only thing that depends on $\theta$ is $g_\theta(z)$ for every $z$. By taking derivatives on our cost function

$$\mathbb{E}_{z \sim p(z)} \left[ \nabla_\theta \log(1 - D^*(g_\theta(z))) \right]$$

$$= \mathbb{E}_{z \sim p(z)} \left[ \nabla_\theta \log \frac{P_{g+\epsilon}(g_\theta(z))}{P_{r+\epsilon}(g_\theta(z)) + P_{g+\epsilon}(g_\theta(z))} \right]$$

$$= \mathbb{E}_{z \sim p(z)} \left[ \nabla_\theta \log P_{g+\epsilon}(g_\theta(z)) - \nabla_\theta \log \left( P_{g+\epsilon}(g_\theta(z)) + P_{r+\epsilon}(g_\theta(z)) \right) \right]$$

$$= \mathbb{E}_{z \sim p(z)} \left[ \frac{\nabla_\theta P_{g+\epsilon}(g_\theta(z))}{P_{g+\epsilon}(g_\theta(z))} - \frac{\nabla_\theta P_{g+\epsilon}(g_\theta(z)) + \nabla_\theta P_{r+\epsilon}(g_\theta(z))}{P_{g+\epsilon}(g_\theta(z)) + P_{r+\epsilon}(g_\theta(z))} \right]$$

$$= \mathbb{E}_{z \sim p(z)} \left[ \frac{1}{P_{g+\epsilon}(g_\theta(z)) + P_{r+\epsilon}(g_\theta(z))} \nabla_\theta [-P_{r+\epsilon}(g_\theta(z))] - \right.$$

$$\left. \frac{1}{P_{g+\epsilon}(g_\theta(z)) + P_{r+\epsilon}(g_\theta(z))} \frac{P_{r+\epsilon}(g_\theta(z))}{P_{g+\epsilon}(g_\theta(z))} \nabla_\theta [-P_{g+\epsilon}(g_\theta(z))] \right]$$

Let the density of $\epsilon$ be $\frac{1}{Z} e^{-\frac{\|x\|^2}{2\sigma^2}}$. We now define

$$a(z) = \frac{1}{2\sigma^2} \frac{1}{P_{g+\epsilon}(g_\theta(z)) + P_{r+\epsilon}(g_\theta(z))}$$

$$b(z) = \frac{1}{2\sigma^2} \frac{1}{P_{g+\epsilon}(g_\theta(z)) + P_{r+\epsilon}(g_\theta(z))} \frac{P_{r+\epsilon}(g_\theta(z))}{P_{g+\epsilon}(g_\theta(z))}$$

Trivially, $a$ and $b$ are positive functions. Since $b = a \frac{P_{r+\epsilon}}{P_{g+\epsilon}}$, we know that $b > a$ if and only if $P_{r+\epsilon} > P_{g+\epsilon}$, and $b < a$ if and only if $P_{r+\epsilon} < P_{g+\epsilon}$ as we wanted. Continuing the proof, we know

$$\mathbb{E}_{z \sim p(z)} \left[ \nabla_\theta \log(1 - D^*(g_\theta(z))) \right]$$

$$= \mathbb{E}_{z \sim p(z)} \left[ 2\sigma^2 a(z) \nabla_\theta [-P_{r+\epsilon}(g_\theta(z))] - 2\sigma^2 b(z) \nabla_\theta [-P_{g+\epsilon}(g_\theta(z))] \right]$$

$$= \mathbb{E}_{z \sim p(z)} \left[ 2\sigma^2 a(z) \int_{\mathcal{M}} -\nabla_\theta \frac{1}{Z} e^{-\frac{\|g_\theta(z) - y\|_2^2}{2\sigma^2}} \, \mathrm{d}\mathbb{P}_r(y) - 2\sigma^2 b(z) \int_{\mathcal{P}} -\nabla_\theta \frac{1}{Z} e^{-\frac{\|g_\theta(z) - y\|_2^2}{2\sigma^2}} \, \mathrm{d}\mathbb{P}_g(y) \right]$$

$$= \mathbb{E}_{z \sim p(z)} \left[ a(z) \int_{\mathcal{M}} \frac{1}{Z} e^{-\frac{\|g_\theta(z) - y\|_2^2}{2\sigma^2}} \nabla_\theta \|g_\theta(z) - y\|^2 \, \mathrm{d}\mathbb{P}_r(y) \right.$$

$$- b(z) \int_{\mathcal{P}} \frac{1}{Z} e^{-\frac{\|g_\theta(z) - y\|_2^2}{2\sigma^2}} \nabla_\theta \|g_\theta(z) - y\|^2 \, \mathrm{d}\mathbb{P}_g(y) \right]$$

$$= \mathbb{E}_{z \sim p(z)} \left[ a(z) \int_{\mathcal{M}} P_\epsilon(g_\theta(z) - y) \nabla_\theta \|g_\theta(z) - y\|^2 \, \mathrm{d}\mathbb{P}_r(y) \right.$$

$$- b(z) \int_{\mathcal{P}} P_\epsilon(g_\theta(z) - y) \nabla_\theta \|g_\theta(z) - y\|^2 \, \mathrm{d}\mathbb{P}_g(y) \right]$$

Finishing the proof. $\qquad \square$

An interesting observation is that if we have two distributions $\mathbb{P}_r$ and $\mathbb{P}_g$ with support on manifolds that are close, the noise terms will make the noisy distributions $\mathbb{P}_{r+\epsilon}$ and $\mathbb{P}_{g+\epsilon}$ almost overlap, and the JSD between them will be small. This is in drastic contrast to the noiseless variants $\mathbb{P}_r$ and $\mathbb{P}_g$, where all the divergences are maxed out, regardless of the closeness of the manifolds. We could argue to use the JSD of the noisy variants to measure a similarity between the original distributions, but this would depend on the amount of noise, and is not an intrinsic measure of $\mathbb{P}_r$ and $\mathbb{P}_g$. Luckily, there are alternatives.

**Definition 3.1.** We recall the definition of the Wasserstein metric $W(P, Q)$ for $P$ and $Q$ two distributions over $\mathcal{X}$. Namely,

$$W(P, Q) = \inf_{\gamma \in \Gamma} \int_{\mathcal{X} \times \mathcal{X}} \|x - y\|_2 d\gamma(x, y)$$

where $\Gamma$ is the set of all possible joints on $\mathcal{X} \times \mathcal{X}$ that have marginals $P$ and $Q$.

The Wasserstein distance also goes by other names, most commonly the transportation metric and the earth mover's distance. This last name is most explicative: it's the minimum cost of transporting the whole probability mass of $P$ from its support to match the probability mass of $Q$ on $Q$'s support. This identification of transporting points from $P$ to $Q$ is done via the coupling $\gamma$. We refer the reader to Villani (2009) for an in-depth explanation of these ideas. It is easy to see now that the Wasserstein metric incorporates the notion of distance (as also seen inside the integral) between the elements in the support of $P$ and the ones in the support of $Q$, and that as the supports of $P$ and $Q$ get closer and closer, the metric will go to 0, inducing as well a notion of distance between manifolds.

Intuitively, as we decrease the noise, $\mathbb{P}_X$ and $\mathbb{P}_{X+\epsilon}$ become more similar. However, it is easy to see again that $JSD(\mathbb{P}_X\|\mathbb{P}_{X+\epsilon})$ is maxed out, regardless of the amount of noise. The following Lemma shows that this is not the case for the Wasserstein metric, and that it goes to 0 smoothly when we decrease the variance of the noise.

**Lemma 4.** *If $\epsilon$ is a random vector with mean 0, then we have*

$$W(\mathbb{P}_X, \mathbb{P}_{X+\epsilon}) \leq V^{\frac{1}{2}}$$

*where $V = \mathbb{E}[\|\epsilon\|_2^2]$ is the variance of $\epsilon$.*

*Proof.* Let $x \sim \mathbb{P}_X$, and $y = x + \epsilon$ with $\epsilon$ independent from $x$. We call $\gamma$ the joint of $(x, y)$, which clearly has marginals $\mathbb{P}_X$ and $\mathbb{P}_{X+\epsilon}$. Therefore,

$$
\begin{aligned}
W(\mathbb{P}_X, \mathbb{P}_{X+\epsilon}) &\leq \int \|x - y\|_2 d\gamma(x, y) \\
&= \mathbb{E}_{x \sim \mathbb{P}_X} \mathbb{E}_{y \sim x+\epsilon}[\|x - y\|_2] \\
&= \mathbb{E}_{x \sim \mathbb{P}_X} \mathbb{E}_{y \sim x+\epsilon}[\|\epsilon\|_2] \\
&= \mathbb{E}_{x \sim \mathbb{P}_X} \mathbb{E}_{\epsilon}[\|\epsilon\|_2] \\
&= \mathbb{E}_{\epsilon}[\|\epsilon\|_2] \\
&\leq \mathbb{E}_{\epsilon}[\|\epsilon\|_2^2]^{\frac{1}{2}} = V^{\frac{1}{2}}
\end{aligned}
$$

where the last inequality was due to Jensen. $\qquad\square$

We now turn to one of our main results. We are interested in studying the distance between $\mathbb{P}_r$ and $\mathbb{P}_g$ without any noise, even when their supports lie on different manifolds, since (for example) the closer these manifolds are, the closer to actual points on the data manifold the samples will be. Furthermore, we eventually want a way to evaluate generative models, regardless of whether they are continuous (as in a VAE) or not (as in a GAN), a problem that has for now been completely unsolved. The next theorem relates the Wasserstein distance of $\mathbb{P}_r$ and $\mathbb{P}_g$, without any noise or modification, to the divergence of $\mathbb{P}_{r+\epsilon}$ and $\mathbb{P}_{g+\epsilon}$, and the variance of the noise. Since $\mathbb{P}_{r+\epsilon}$ and $\mathbb{P}_{g+\epsilon}$ are continuous distributions, this divergence is a sensible estimate, which can even be attempted to minimize, since a discriminator trained on those distributions will approximate the JSD between them, and provide smooth gradients as per Corolary 3.2.

**Theorem 3.3.** *Let $\mathbb{P}_r$ and $\mathbb{P}_g$ be any two distributions, and $\epsilon$ be a random vector with mean 0 and variance $V$. If $\mathbb{P}_{r+\epsilon}$ and $\mathbb{P}_{g+\epsilon}$ have support contained on a ball of diameter $C$, then* [6]

$$W(\mathbb{P}_r, \mathbb{P}_g) \leq 2V^{\frac{1}{2}} + 2C\sqrt{JSD(\mathbb{P}_{r+\epsilon}\|\mathbb{P}_{g+\epsilon})} \tag{6}$$

*Proof.*

$$
\begin{aligned}
W(\mathbb{P}_r, \mathbb{P}_g) &\leq W(\mathbb{P}_r, \mathbb{P}_{r+\epsilon}) + W(\mathbb{P}_{r+\epsilon}, \mathbb{P}_{g+\epsilon}) + W(\mathbb{P}_{g+\epsilon}, \mathbb{P}_g) \\
&\leq 2V^{\frac{1}{2}} + W(\mathbb{P}_{r+\epsilon}, \mathbb{P}_{g+\epsilon}) \\
&\leq 2V^{\frac{1}{2}} + C\delta(\mathbb{P}_{r+\epsilon}, \mathbb{P}_{g+\epsilon}) \\
&\leq 2V^{\frac{1}{2}} + C\left(\delta(\mathbb{P}_{r+\epsilon}, \mathbb{P}_m) + \delta(\mathbb{P}_{g+\epsilon}, \mathbb{P}_m)\right) \\
&\leq 2V^{\frac{1}{2}} + C\left(\sqrt{\frac{1}{2}KL(\mathbb{P}_{r+\epsilon}\|\mathbb{P}_m)} + \sqrt{\frac{1}{2}KL(\mathbb{P}_{g+\epsilon}\|\mathbb{P}_m)}\right) \\
&\leq 2V^{\frac{1}{2}} + 2C\sqrt{JSD(\mathbb{P}_{r+\epsilon}\|\mathbb{P}_{g+\epsilon})}
\end{aligned}
$$

We first used the Lemma 4 to bound everything but the middle term as a function of $V$. After that, we followed by the fact that $W(P,Q) \leq C\delta(P,Q)$ wih $\delta$ the total variation, which is a popular Lemma arizing from the Kantorovich-Rubinstein duality. After that, we used the triangular inequality on $\delta$ and $\mathbb{P}_m$ the mixture distribution between $\mathbb{P}_{g+\epsilon}$ and $\mathbb{P}_{r+\epsilon}$. Finally, we used Pinsker's inequality and later the fact that each individual $KL$ is only one of the non-negative sumands of the $JSD$.  $\square$

Theorem 3.3 points us to an interesting idea. The two terms in equation (6) can be controlled. The first term can be decreased by annealing the noise, and the second term can be minimized by a GAN when the discriminator is trained on the noisy inputs, since it will be approximating the JSD between the two continuous distributions. One great advantage of this is that we no longer have to worry about training schedules. Because of the noise, we can train the discriminator till optimality without any problems and get smooth interpretable gradients by Corollary 3.2. All this while still minimizing the distance between $\mathbb{P}_r$ and $\mathbb{P}_g$, the two noiseless distributions we in the end care about.

### ACKNOWLEDGMENTS

The first author would like to especially thank Luis Scoccola for help with the proof of Lemma 1.

The authors would also like to thank Ishmael Belghazi, Yoshua Bengio, Gerry Che, Soumith Chintala, Caglar Gulcehre, Daniel Jiwoong Im, Alex Lamb, Luis Scoccola, Pablo Sprechmann, Arthur Szlam, Jake Zhao for insightful comments and advice.

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

## A   PROOFS OF THINGS

*Proof of Lemma 1.* We first consider the case where the nonlinearities are rectifiers or leaky rectifiers of the form $\sigma(x) = \mathbb{1}[x < 0]c_1 x + \mathbb{1}[x \geq 0]c_2 x$ for some $c_1, c_2 \in \mathbb{R}$. In this case, $g(z) = \mathbf{D}_n \mathbf{W}_n \ldots \mathbf{D}_1 \mathbf{W}_1 z$, where $\mathbf{W}_i$ are affine transformations and $\mathbf{D}_i$ are some diagonal matrices dependent on $z$ that have diagonal entries $c_1$ or $c_2$. If we consider $\mathcal{D}$ to be the (finite) set of all diagonal matrices with diagonal entries $c_1$ or $c_2$, then $g(\mathcal{Z}) \subseteq \bigcup_{D_i \in \mathcal{D}} \mathbf{D}_n \mathbf{W}_n \ldots \mathbf{D}_1 \mathbf{W}_1 \mathcal{Z}$, which is a finite union of linear manifolds.

The proof for the second case is technical and slightly more involved. When $\sigma$ is a pointwise smooth strictly increasing nonlinearity, then applying it vectorwise it's a diffeomorphism to its image. Therefore, it sends a countable union of manifolds of dimension $d$ to a countable union of manifolds of dimension $d$. If we can prove the same thing for affine transformations we will be finished, since $g(\mathcal{Z})$ is just a composition of these applied to a $\dim \mathcal{Z}$ dimensional manifold. Of course, it suffices to prove that an affine transformation sends a manifold to a countable union of manifolds without increasing dimension, since a countable union of countable unions is still a countable union. Furthermore, we only need to show this for linear transformations, since applying a bias term is a diffeomorphism.

Let $\mathbf{W} \in \mathbb{R}^{n \times m}$ be a matrix. Note that by the singular value decomposition, $\mathbf{W} = \mathbf{U}\mathbf{\Sigma}\mathbf{V}$, where $\mathbf{\Sigma}$ is a square diagonal matrix with diagonal positive entries and $\mathbf{U}, \mathbf{V}$ are compositions of changes of basis, inclusions (meaning adding 0s to new coordinates) and projections to a subset of the coordinates. Multiplying by $\Sigma$ and applying a change of basis are diffeomorphisms, and adding 0s to new coordinates is a manifold embedding, so we only need to prove our statement for projections onto a subset of the coordinates. Let $\pi : \mathbb{R}^{n+k} \to \mathbb{R}^n$, where $\pi(x_1, \ldots, x_{n+k}) = (x_1, \ldots, x_n)$ be our projection and $\mathcal{M} \subseteq \mathbb{R}^{n+k}$ our $d$-dimensional manifold. If $n \leq d$, we are done since the image of $\pi$ is contained in all $\mathbb{R}^n$, a manifold with at most dimension $d$. We now turn to the case where $n > d$. Let $\pi_i(x) = x_i$ be the projection onto the $i$-th coordinate. If $x$ is a critical point of $\pi$, since the coordinates of $\pi$ are independent, then $x$ has to be a critical point of a $\pi_i$. By a consequence of the Morse Lemma, the critical points of $\pi_i$ are isolated, and therefore so are the ones of $\pi$, meaning that there is at most a countable number of them. Since $\pi$ maps the non-critical points onto a $d$ dimensional manifold (because it acts as an embedding) and the countable number of critical points into a countable number of points (or 0 dimensional manifolds), the proof is finished. $\qquad\square$

*Proof of Lemma 2.* For now we assume that $\mathcal{M}$ and $\mathcal{P}$ are without boundary. If $\dim \mathcal{M} + \dim \mathcal{P} \geq d$ it is known that under arbitrarilly small perturbations defined as the ones in the statement of this Lemma, the two dimensions will intersect only transversally with probability 1 by the General Position Lemma. If $\dim \mathcal{M} + \dim \mathcal{P} < d$, we will show that with probability 1, $\mathcal{M} + \eta$ and $\mathcal{P} + \eta'$ will not intersect, thereby getting our desired result. Let us then assume $\dim \mathcal{M} + \dim \mathcal{P} < d$. Note that $\hat{\mathcal{M}} \cap \hat{\mathcal{P}} \neq \emptyset$ if and only if there are $x \in \mathcal{M}, y \in \mathcal{P}$ such that $x + \eta = y + \eta'$, or equivalently $x - y = \eta' - \eta$. Therefore, $\hat{\mathcal{M}}$ and $\hat{\mathcal{P}}$ intersect if and only if $\eta' - \eta \in \mathcal{M} - \mathcal{P}$. Since $\eta, \eta'$ are independent continuous random variables, the difference is also continuous. If $\mathcal{M} - \mathcal{P}$ has measure 0 in $\mathbb{R}^d$ then $\mathbb{P}(\eta' - \eta \in \mathcal{M} - \mathcal{P}) = 0$, concluding the proof. We will therefore show that $\mathcal{M} - \mathcal{P}$ has measure 0. Let $f : \mathcal{M} \times \mathcal{P} \to \mathbb{R}^d$ be $f(x, y) = x - y$. If $m$ and $p$ are the dimensions of $\mathcal{M}$ and $\mathcal{P}$, then $f$ is a smooth function between an $m + p$-dimensional manifold and a $d$ dimensional one. Clearly, the image of $f$ is $\mathcal{M} - \mathcal{P}$. Therefore,

$$\mathcal{M} - \mathcal{P} = f(\{z \in \mathcal{M} \times \mathcal{P} | \text{rank}(d_z f) < m + p\})$$
$$\cup f(\{z \in \mathcal{M} \times \mathcal{P} | \text{rank}(d_z f) = m + p\})$$

The first set is the image of the critical points, namely the critical values. By Sard's Lemma, this set has measure 0. Let's call $A = \{z \in \mathcal{M} \times \mathcal{P} | \text{rank}(d_z f) = m + p\}$. Let $z$ be an element of $A$. By the inverse function theorem, there is a neighbourhood $U_z \subseteq \mathcal{M} \times \mathcal{P}$ of $z$ such that $f|_{U_z}$ is an embedding. Since every manifold has a countable topological basis, we can cover $A$ by countable sets $U_{z_n}$, where $n \in \mathbb{N}$. We will just note them by $U_n$. Since $f|_{U_n}$ is an embedding, $f(U_n)$ is an $m + p$-dimensional manifold, and since $m + p < d$, this set has measure 0 in $\mathbb{R}^d$. Now, $f(A) = \bigcup_{n \in \mathbb{N}} f(U_n)$, which therefore has measure 0 in $\mathbb{R}^d$, finishing the proof of the boundary free case.

Now we consider the case where $\mathcal{M}$ and $\mathcal{P}$ are manifolds with boundary. By a simple union bound,

$$\mathbb{P}_{\eta,\eta'}(\tilde{\mathcal{M}} \text{ perfectly aligns with } \tilde{\mathcal{P}}) \leq \mathbb{P}_{\eta,\eta'}(\text{Int } \tilde{\mathcal{M}} \text{ perfectly aligns with Int } \tilde{\mathcal{P}})$$
$$+ \mathbb{P}_{\eta,\eta'}(\text{Int } \tilde{\mathcal{M}} \text{ perfectly aligns with } \partial\tilde{\mathcal{P}})$$
$$+ \mathbb{P}_{\eta,\eta'}(\partial\tilde{\mathcal{M}} \text{ perfectly aligns with Int } \tilde{\mathcal{P}})$$
$$+ \mathbb{P}_{\eta,\eta'}(\partial\tilde{\mathcal{M}} \text{ perfectly aligns with } \partial\tilde{\mathcal{P}})$$
$$= 0$$

where the last equality arizes when combining the facts that $\text{Int} \, \tilde{\mathcal{M}} = \eta + \text{Int } \mathcal{M} = \text{Int } (\eta + \mathcal{M}) = \text{Int } \tilde{\mathcal{M}}$ (and analogously for the boundary and $\mathcal{P}$), that the boundary and interiors of $\mathcal{M}$ and $\mathcal{P}$ are boundary free regular submanifolds of $\mathbb{R}^d$ without full dimension, and then applying the boundary free case of the proof. $\qquad \square$

*Proof of Lemma 3.* Let $m = \dim \mathcal{M}$ and $p = \dim \mathcal{P}$. We again consider first the case where $\mathcal{M}$ and $\mathcal{P}$ are manifolds without boundary. If $m + p < d$, then $\mathcal{L} = \emptyset$ so the statement is obviously true. If $m + p \geq d$, then $\mathcal{M}$ and $\mathcal{P}$ intersect transversally. This implies that $\mathcal{L}$ is a manifold of dimension $m + p - d < m, p$. Since $\mathcal{L}$ is a submanifold of both $\mathcal{M}$ and $\mathcal{P}$ that has lower dimension, it has measure 0 on both of them.

We now tackle the case where $\mathcal{M}$ and $\mathcal{P}$ have boundaries. Let us remember that $\mathcal{M} = \text{Int } \mathcal{M} \cup \partial \mathcal{M}$ and the union is disjoint (and analogously for $\mathcal{P}$). By using elementary properties of sets, we can trivially see that

$$\mathcal{L} = \mathcal{M} \cap \mathcal{P} = (\text{Int } \mathcal{M} \cap \text{Int } \mathcal{P}) \cup (\text{Int } \mathcal{M} \cap \partial\mathcal{P}) \cup (\partial\mathcal{M} \cap \text{Int } \mathcal{P}) \cup (\partial\mathcal{M} \cap \partial\mathcal{P})$$

where the unions are disjoint. This is the disjoint union of 4 strictly lower dimensional manifolds, by using the first part of the proof. Since each one of these intersections has measure 0 on either the interior or boundary of $\mathcal{M}$ (again, by the first part of the proof), and interior and boundary are contained in $\mathcal{M}$, each one of the four intersections has measure 0 in $\mathcal{M}$. Analogously, they have measure 0 in $\mathcal{P}$, and by a simple union bound we see that $\mathcal{L}$ has measure 0 in $\mathcal{M}$ and $\mathcal{P}$ finishing the remaining case of the proof. $\qquad \square$

*Proof of Theorem 3.1.* We first need to show that $\mathbb{P}_{X+\epsilon}$ is absolutely continuous. Let $A$ be a Borel set with Lebesgue measure 0. Then, by the fact that $\epsilon$ and $X$ are independent, we know by Fubini

$$\mathbb{P}_{X+\epsilon}(A) = \int_{\mathbb{R}^d} \mathbb{P}_\epsilon(A - x) \, \mathrm{d}\mathbb{P}_X(x)$$
$$= \int_{\mathbb{R}^d} 0 \, \mathrm{d}\mathbb{P}_X(x) = 0$$

Where we used the fact that if $A$ has Lebesgue measure zero, then so does $A - x$ and since $\mathbb{P}_\epsilon$ is absolutely continuous, $\mathbb{P}_\epsilon(A - x) = 0$.

Now we calculate the density of $\mathbb{P}_{X+\epsilon}$. Again, by using the independence of $X$ and $\epsilon$, for any Borel set $B$ we know

$$\mathbb{P}_{X+\epsilon}(B) = \int_{\mathbb{R}^d} \mathbb{P}_\epsilon(B - y) \, \mathrm{d}\mathbb{P}_X(y)$$
$$= \mathbb{E}_{y \sim \mathbb{P}_X} \left[ \mathbb{P}_\epsilon(B - y) \right]$$
$$= \mathbb{E}_{y \sim \mathbb{P}_x} \left[ \int_{B-y} P_\epsilon(x) dx \right]$$
$$= \mathbb{E}_{y \sim \mathbb{P}_x} \left[ \int_B P_\epsilon(x - y) dx \right]$$
$$= \int_B \mathbb{E}_{y \sim \mathbb{P}_x} \left[ P_\epsilon(x - y) \right] dx$$

Therefore, $\mathbb{P}_{X+\epsilon}(B) = \int_B P_{X+\epsilon}(x) dx$ for our proposed $P_{X+\epsilon}$ and all Borel sets $B$. By the uniqueness of the Radon-Nikodym theorem, this implies the proposed $P_{X+\epsilon}$ is the density of $\mathbb{P}_{X+\epsilon}$. The equivalence of the formula changing the expectation for $\int_{\mathcal{M}} \mathbb{P}_X$ is trivial by the definition of expectation and the fact that the support of $\mathbb{P}_X$ lies on $\mathcal{M}$. $\qquad \square$

## B    FURTHER CLARIFICATIONS

In this appendix we further explain some of the terms and ideas mentioned in the paper, which due to space constrains, and to keep the flow of the paper, couldn't be extremely developed in the main text. Some of these have to do with notation, others with technical elements of the proofs. On the latter case, we try to convey more intuition than we previously could. We present these clarifications in a very informal fashion in the following item list.

- There are two different but very related properties a random variable can have. A random variable $X$ is said to be continuous if $P(X = x) = 0$ for all single points $x \in \mathcal{X}$. Note that a random variable concentrated on a low dimensional manifold such as a plane can have this property. However, an absolutely continuous random variable has the following property: if a set $A$ has Lebesgue measure 0, then $P(X \in A) = 0$. Since points have measure 0 with the Lebesgue measure, absolute continuity implies continuity. A random variable that's supported on a low dimensional manifold therefore will not be absolutely continuous: let $\mathcal{M}$ a low dimensional manifold be the support of $X$. Since a low dimensional manifold has 0 Lebesgue measure, this would imply $P(X \in M) = 0$, which is an absurd since $\mathcal{M}$ was the support of $X$. The property of $X$ being absolutely continuous can be shown to be equivalent to $X$ having a density: the existence of a function $f : \mathcal{X} \to \mathbb{R}$ such that $P(X \in A) = \int_A f(x) \, \mathrm{d}x$ (this is a consequence of the Radon-Nikodym theorem).
  The annoying part is that in everyday paper writing when we talk about continuous random variables, we omit the "absolutely" word to keep the text concise and actually talk about absolutely continuous random variables (ones that have a density), this is done through almost all sciences and throughout mathematics as well, annoying as it is. However we made the clarification in here since it's relevant to our paper not to mistake the two terms.

- The notation $\mathbb{P}_r[D(x) = 1] = 1$ is the abbreviation of $\mathbb{P}_r[\{x \in \mathcal{X} : D(x) = 1\}] = 1$ for a measure $\mathbb{P}_r$. Another way of expressing this more formally is $\mathbb{P}_r[D^{-1}(1)] = 1$.

- In the proof of Theorem 2.1, the distance between sets $d(A, B)$ is defined as the usual distance between sets in a metric space

$$d(A, B) = \inf_{x \in A, y \in B} d(x, y)$$

  where $d(x, y)$ is the distance between points (in our case the Euclidean distance).

- Note that not everything that's outside of the support of $\mathbb{P}_r$ has to be a generated image. Generated images are only things that lie in the support of $\mathbb{P}_g$, and there are things that don't need to be in the support of either $\mathbb{P}_r$ or $\mathbb{P}_g$ (these could be places where $0 < D < 1$ for example). This is because the discriminator is not trained to discriminate $\mathbb{P}_r$ from all things that are not $\mathbb{P}_r$, but to distinguish $\mathbb{P}_r$ from $\mathbb{P}_g$. Points that don't lie in the support of $\mathbb{P}_r$ or $\mathbb{P}_g$ are not important to the performance of the discriminator (as is easily evidenced in its cost). Why we define accuracy 1 as is done in the text is to avoid the identification of a single 'tight' support, since this typically leads to problems (if I take a measure 0 set from any support it still is the support of the distribution). In the end, what we aim for is:
  - We want $D(x) = 1$ with probability 1 when $x \sim \mathbb{P}_r$.
  - We want $D(x) = 0$ with probability 1 when $x \sim \mathbb{P}_g$.
  - Whatever happens elsewhere is irrelevant (as it is also reflected by the cost of the discriminator)

- We say that a discriminator $D^*$ is optimal for $g_\theta$ (or its corresponding $\mathbb{P}_g$) if for all measurable functions $D : \mathcal{X} \to [0, 1]$ we have

$$L(D^*, g_\theta) \geq L(D, g_\theta)$$

  for $L$ defined as in equation (1).

