# Peer review of "Towards Principled Methods for Training Generative Adversarial Networks"

_ICLR 2017 — accepted_

[Public Comment · Antonia Creswell · 16 Dec 2016]
**Questions and inconsistencies.**

This paper makes many valuable contributions and explains many of the issues related to training GANs; I feel it will be an essential platform for future work. However I have the following questions and found the following inconsistencies.

1) 
	“If n <= d, we are done since the image of π is contained in all Rn, a manifold with at 
	most dimension d. We now turn to the case where d > n.”

Is the case  d>n  not contained in the first,  n<=d ? Should this be n>d?

2) The perfect discriminator theorems: 

	“We say that a discriminator D : X → [0, 1] has accuracy 1 if it takes the value 1 on a set that 
	contains the support of P_r  and value 0 on a set that contains the support of P_g .”

If a set S_g or S_r contains the support of P_g or P_r, this suggests that it may contain the support but may also contain regions outside of the support.  Are you suggesting that for x in S_r that are outside the support of P_r, D(x)=1? However, surely regions outside the support of P_r should correspond to generated images, so D(x) should be 0.

b) Following from a), if a set S_g (or S_r) contained regions outside the support (as well as the support) then S_g (or S_r)  would contain the support of P_g (or P_r) but may also contain regions in the support of P_r (or P_g).

3) 
		P_r[D(x) = 1] = 1 and P_g[D(x) = 0] = 1 

a) This notation is not clear, do you mean:
	P_r(x)=1 for all x in {x : D(x)=1} and P_g(x)=1 for all x in {x : D(x)=0}

4) Theorem 2.1

a) It is not clear what distance measure you are using between sets
b) Where does the delta/3 come from? Is this arbitrary?

5) Theorem 2.2
a) What is P^c?
b) Is the ball B(x,e_x) defined for all x in M\L? This is not explicit.

If defined for all x in M\L:
c) For x on the boundary of M\L and P\L is e_x=0? Otherwise, is it possible that for x on the boundary of M\L or P\L , M_hat and P_hat intersect with P\L or M\L respectively? 

Finally:
d) M_hat is a superset of M\L, how can you say D*(x)=1 for all x in M_hat, when M_hat may include regions outside the support of M?

9) Corollary 2.1; 
	"Under the same assumptions of Theorem 1.3 "
a) What is Theorem 1.3?

11) Theorem 2.5 
a) Could this be explicitly linked to the collapsing generator problem talked about by [1] and [2].  Citations might make the link clearer. 

12) Theorem 2.6 
a) Why do you assume that the distribution of difference between D and D* is white noise? This suggests that most of the time D is optimal, from the rest of the paper this is believable, however if this is what you are assuming, this could be made more explicit. 
b) Similar question for difference in gradient, though since grad D* is zero most of the time, that would suggest that grad D is zero most of the time. Is this true?
c) extra bracket in final line of proof

16) 
       “This is in drastic contrast to the noiseless variants P_g  and P_g,”
a) repeated P_g

17) 
         “again that JSD(P_x,P_{x+e} is maxed out,”
a) missing bracket and inconsistent notation, JSD(P_x||P_{x+e})
b) “maxed out“ is a colloquialism

[1] Salimans, Tim, et al. "Improved techniques for training gans." arXiv preprint arXiv:1606.03498 (2016).
[2]Odena, Augustus, Christopher Olah, and Jonathon Shlens. "Conditional Image Synthesis With Auxiliary Classifier GANs." arXiv preprint arXiv:1610.09585 (2016).

[Public Comment · (anonymous) · rating 10 · confidence 5 · 17 Dec 2016]
**very solid contribution to the theoretical understanding of GANS**

This is a strong submission regarding one of the most important and recently introduced methods in neural networks - generative adversarial networks. The authors analyze theoretically the convergence of GANs and discuss the stability of GANs. Both are very important. To the best of my knowledge, this is one of the first theoretical papers about GANs and the paper, contrary to most of the submissions in the field, actually provides deep theoretical insight into this architecture. The stability issues regarding GANs are extremely important since the first proposed versions of GANs architecture were very unstable and did not work well in practice. Theorems 2.4-2.6 are novel and introduces mathematical techniques are interesting. I have some technical questions regarding the proof of Theorem 2.5 but these are pretty minor.

[Official Review · AnonReviewer2 · rating 10 · confidence 5 · 19 Dec 2016]
**very interesting submission**
originality 2 · impact 3

This is a strong submission regarding one of the most important and recently introduced methods in neural networks - generative adversarial networks. The authors analyze theoretically the convergence of GANs and discuss the stability of GANs. Both are very important. To the best of my knowledge, this is one of the first theoretical papers about GANs and the paper, contrary to most of the submissions in the field, actually provides deep theoretical insight into this architecture. The stability issues regarding GANs are extremely important since the first proposed versions of GANs architecture were very unstable and did not work well in practice. Theorems 2.4-2.6 are novel and introduces mathematical techniques are interesting. I have some technical questions regarding the proof of Theorem 2.5 but these are pretty minor.

[Official Review · AnonReviewer1 · rating 7 · confidence 4 · 19 Dec 2016]
**review of ``TOWARDS PRINCIPLED METHODS FOR TRAINING GENERATIVE ADVERSARIAL NETWORKS''**
soundness 4 · clarity 4 · meaningful comparison 4 · recommendation (unofficial) 2

SUMMARY 
This paper addresses important questions about the difficulties in training generative adversarial networks. It discusses consequences of using an asymmetric divergence function and sources of instability in training GANs. Then it proposes an alternative using a smoothening approach. 

PROS 
Theory, good questions, nice answers. 
Makes an interesting use of concepts form analysis and differential topology. 
Proposes avenues to avoid instability in GANs. 

CONS 
A bit too long, technical. Some parts and consequences still need to be further developed (which is perfectly fine for future work). 

MINOR COMMENTS

- Section 2.1 Maybe shorten this section a bit. E.g., move all proofs to the appendix. 

- Section 3 provides a nice, intuitive, simple solution. 

- On page 2 second bullet. This also means that P_g is smaller than the data distribution in some other x, which in turn will make the KL divergence non zero. 

- On page 2, ``for not generating plausibly looking pictures'' should be ``for generating not plausibly looking pictures''.  

- Lemma 1 would also hold in more generality. 

- Theorem 2.1 seems to be basic analysis. (In other words, a reference could spare the proof). 

- In Theorem 2.4, it would be good to remind the reader about p(z). 

- Lemma 2 seems to be basic analysis. (In other words, a reference could spare the proof). 
Specify the domain of the random variables. 

- relly - > rely 

- Theorem 2.2 the closed manifolds have boundary or not? (already in the questions)

- Corollary 2.1, ``assumptions of Theorem 1.3''. I could not find Theorem 1.3. 

- Theorem 2.5 ``Therefore'' -> `Then'? 

- Theorem 2.6 ``Is a... '' -> `is a' ? 

- The number of the theorems is confusing.

[Official Review · AnonReviewer3 · rating 8 · confidence 3 · 20 Dec 2016]
**good submission**
originality 2 · appropriateness 2 · meaningful comparison 3

This paper makes a valuable contribution to provide a more clear understanding of generative adversarial network (GAN) training procedure. 

With the new insight of the training dynamics of GAN, as well as its variant, the authors reveal the reason that why the gradient is either vanishing in original GAN or unstable in its variant. More importantly, they also provide a way to avoid such difficulties by introducing perturbation. I believe this paper will inspire more principled research in this direction. 

I am very interested in the perturbation trick to avoid the gradient instability and vanishment. In fact, this is quite related to dropout trick in where the perturbation can be viewed as Bernoulli distribution. It will be great if the connection can be discussed.  Besides the theoretical analysis, is there any empirical study to justify this trick? Could you please add some experiments like Fig 2 and 3 for the perturbated GAN for comparison?

[Author Response · Martin Arjovsky · 24 Dec 2016]
**Revision**

Hi! We would first like to thank the reviewers for your comments. We will aim to make all the suggestions fit into a revision. We will update the paper shortly and provide individual responses to the reviews :)

Best!
Martin

[Public Comment · Sander Dieleman · 28 Dec 2016]
**last paragraph**

I've always considered myself something of a GAN sceptic, so I'm very excited to see so many papers popping up that try to analyse and address their shortcomings. There are many ICLR submissions on this topic but I found this one a particularly enlightening read, nice work!

The last paragraph of the paper is perhaps the most interesting, as it seems to propose an alternative training method that would solve most of the problems: the gradients no longer vanish when the discriminator is trained to convergence, which eliminates the careful balancing act that GAN training currently requires.

But unfortunately the paper ends there, with no discussion of how this would be implemented in practice and no experiments to demonstrate the proposed solution. Is this work that is currently underway? It seems like an excellent idea so it's strange that only one paragraph of the paper is dedicated to it (although admittedly the paper is already quite long).

[Public Comment · Michael Mathieu · 03 Jan 2017]
**Very good paper**

I think this paper is a very good step towards interpreting and understanding GANs. While some experiments have been (often successfully) done with adding noise, there was no theoretical reason, mainly intuition and hand waving.
This paper clearly states the reason why these methods work, by pointing a fundamental pitfall in GANs that was overlooked so far.
I believe that this is a crucial step towards making useful, large scale GANs.

Some notes and typos (sorry if it was already addressed in previous comments):
- First paragraph of section 2, the cost part is quite unclear (it’s the negative of what was described earlier (after equation 2), and the words "maximizing" and "minimizing" are mixed).
- Definition 2.1: what is the tangent space of the whole space around a point? Isn’t it just the point? I think you mean TxM + TxP = F . Also, I thin the + should be a (+) (vector space sum)
- Bottom of page 4, before Lemma 2: its -> their (intersection)
- Maybe I'm not familiar enough with the vocabulary, but I did not quite understand what you meant by "non continuous distribution", despite the footnote. As a consequence, it took me until much later in the paper to fully understand the link between non-continuous and low-dimensional manifold.
- There is a numbering problem. In particular, theorem 2.6 refers to theorems 1.1 and 1.2, which don’t exist.
- Theorem 2.6: I think the definition and role of epsilon could be clarified.
- Corollary 3.2: missing a tilde, g -> g_\theta
- Paragraph above lemma 4: missing ‘)’ after JSD(...

[Author Response · Martin Arjovsky · 12 Jan 2017]
**New version**

Hi! We would like to comment that we added a revision with the following changes at its core:

- We extended the proofs and definitions in section 2 to work with manifolds with boundary.

- We included a one page Appendix B with further clarifications for the things that were suggested in the comments, such as a small comment on continuity vs absolute continuity of random variables, and how it is relevant to our paper.

- We did some minor rewriting to take into account the suggestions from the reviewers and commenters.

- We fixed all the typos :)

Best!
Martin

[Reviewer Comment · AnonReviewer2 · 20 Jan 2017]
**final evaluation - strong submission**
originality 2 · impact 3

Very good paper, I hope it will be accepted. I keep my original evaluation.

[Public Comment · Arnaud Sors · 27 Jan 2017]
**high quality paper**

Hi!
I think this is an outstanding paper. 
It has greatly helped me towards understanding where failure modes of GANs come from, and I think it will also have great practical implications on how to train GANs. 

Following your ideas, I have played with added noise on GANs. I am able to get the generator to yield 'good looking' samples in much less G iterations than in the standard setting. Also, it appears that 'successful' training is a LOT less sensitive to the setting of hyperparameters. For example I am able to use 10 times higher learning rates, forget about gradient clipping, etc... so this is great! I think this paper paves the way to further research on the following points:
- how to choose noise variance and schedule its decrease over training ?
- what does 'training D to convergence' mean ? In practice it seems that training D 'more than G' but only a few steps is already sufficient. Can we get a theoretical understanding of this...

Also, in my (quick and dirty) first experiments, using instance noise helped training in less G iterations, but the mode dropping problem remained. In my understanding, your theoretical analysis demonstrates that the addition of noise helps D provide gradients to G so that at anytime, G is able to escape its current modes. However, and due to the fact that D outputs are calculated from single examples there is no 'coverage guarantee', for example G could keep switching between different modes. Is this right ? Minibatch discrimination appears to be a good first way to answer this, although not 100% satisfactory because the metric is combinatorial... What is your view on these practical considerations ?

Many thanks !

Arnaud Sors

[Public Comment · (anonymous) · 02 Feb 2017]
**Theorem 2.1**

I am a bit confused with your proof of Theorem 2.1: shouldn't it be that the Urysohn's lemma imply that D* is 1 on M and 0 on P (not on M^{hat} and P^{hat})?
Also small typo in the formulation: discrimator -> discriminator.

[Final Decision · Program Chairs · 06 Feb 2017]
**ICLR committee final decision**

The paper provides a detailed analysis of the instability issues surrounding the training of GANs. They demonstrate how perturbations can help with improving stability. Given the popularity of GANs, this paper is expected to have a significant impact.